# HIV testing uptake and prevalence among hospitalized older adults in Togo: A cross-sectional study

**Fifonsi Adjidossi Gbeasor-Komlanvi** [1,2]*, **Wendpouire Ida Carine Zida-Compaore**[2], **Arnold Junior Sadio**[1,2], **Martin Kouame Tchankoni** [2], **Balakiyem Magnim Kadangha**[2], **Mounerou Salou**[3], **Anoumou Claver Dagnra**[3,4], **Didier Koumavi Ekouevi**[1,2,5]

1 Département de Santé Publique, Faculté des Sciences de la Santé, Université de Lomé, Lomé, Togo, 2 Centre Africain de Recherche en Epidémiologie et en Santé Publique, Lomé, Togo, 3 Laboratoire de Biologie Moléculaire et d'Immunologie, Université de Lomé, Lomé, Togo, 4 Programme National de Lutte contre le Sida et les Infections Sexuellement Transmissibles, Lomé, Togo, 5 INSERM U1219 Bordeaux Population Health Research, ISPED, Université de Bordeaux, Bordeaux, France

* fifonsi.gbeasor@gmail.com

**Data Availability Statement:** All relevant data are within the manuscript and its Supporting Information files.

## Abstract

### Objectives

This study aimed to describe HIV testing uptake, as well as HIV prevalence and its associated factors among older adults aged ≥50 years in health facilities in Togo.

### Methods

A cross-sectional study was carried out from February 2018 to June 2019 among hospitalized older adults aged ≥50 years in tertiary and secondary hospitals in Togo. HIV testing was performed according to the national algorithm. Socio-demographic data and HIV testing history were collected using a standardized questionnaire.

### Results

A total of 619 patients (43.9% female) of median age 61 years, (IQR: 55–70) were recruited and offered HIV testing. Among them, 25.7% had never previously tested for HIV. In total, 91.6% (567/619) accepted HIV testing while 8.4% (52/619) refused to be tested. Of those who tested, forty patients were HIV positive, yielding a prevalence of 7.1%. Twenty-three patients (57.5%) were newly diagnosed with HIV infection. In multivariable analysis, two factors were associated with HIV infection: living alone (aOR = 5.83; 95%CI = [2.26–14.53]) and being <60 years (aOR = 3.12; 95%CI = [1.51–6.66]).

### Conclusion

The majority of older adults in this study accepted testing for HIV and almost three in five HIV positive older adults were newly diagnosed with HIV as a result of this testing. There is an urgent need to integrate older adults into responses to the HIV epidemic and to strengthen targeted prevention care and treatment in this population.

**Funding:** The author(s) received no specific funding for this work.

**Competing interests:** The authors have declared that no competing interests exist.

**Abbreviations:** 95% CI, 95% confidence interval; aOR, adjusted Odds Ratio; IQR, interquartile range; MD, missing data; PICT, provider-initiated testing and counselling; PLWH, people living with HIV; **PLWH50**+, people living with HIV aged 50 years and older; SSA, Sub-Saharan Africa; VCT, voluntary counselling and testing.

## Introduction

According to the Joint United Nations Programme on HIV/AIDS (UNAIDS), the number of people living with HIV aged 50 and older (PLWH50+) has increased in all regions of the world [1]. Between 2000 and 2019, the number of PLWH50+ has more than doubled in West and Central Africa, while it has quadrupled in Eastern and Southern Africa; in Western and Central Europe and North America, it showed a ten-fold increase [2]. Globally, the proportion of PLWH50+ increased from 8% of all people living with HIV in 2000 to almost 21% in 2019 [2]. In 2019, there were 7.9 million PLWH50+ worldwide [2].

Large-scale access to antiretroviral therapy (ART) has greatly contributed to the increase in the number of older people living with HIV (PLWH). Early initiation of ART has led to a decline in AIDS-related deaths by 43% since 2003 and nowadays, PLWH who start ART could expect to live as long as an HIV-negative person of the same age [3, 4]. Besides large-scale access to ART, the ageing of the population in all regions of the world has led to new infections being reported among older adults. In the United States (US), around one in six new HIV diagnoses occurred among people aged ≥50 years in 2017 [5], while an analysis of surveillance data from 31 European countries reported a significant increase in the notification rate of new diagnoses among older adults between 2004–2015 [6].

The 'greying of the HIV epidemic' raises important public health issues including low rates of HIV testing which have been reported among older adults. In a study conducted among 12,366 people aged ≥50 years in the US, only 25.4% reported lifetime testing for HIV, and of those who had been tested, 69.7% reported having tested more than five years prior to survey [7]. With regards to new infections, older adults are more likely to be diagnosed at a late stage in the course of the HIV disease than younger people [6, 8]. Late diagnosis among older adults is of great concern as it is associated with adverse outcomes such as rapid progression to AIDS, presence of ageing-related comorbidities and high mortality rate [9–11]. To improve early detection of HIV infection, especially among populations otherwise unlikely to obtain an HIV test like older adults, implementation of routine HIV testing is the best strategy and it is recommended for everyone between the ages of 13 and 64 in the US [9, 12]. However, less than two older adults in five in the US who tested for HIV were offered HIV testing during routine medical check-up [7].

In Sub-Saharan Africa (SSA) which bears the heaviest burden of the HIV epidemic, evidence on older PLWH is beginning to emerge [13–20]. Estimates of HIV prevalence among older adults aged ≥50 years have been reported in Tanzania where it ranged from 1.7% [17] to 6.7% [18, 19], in Botswana (the rate was 21.9% in females and 27.8% in males [15]), whilst in South Africa, it was 20% [20]. HIV testing among older adults remains a challenging issue in SSA. Although various HIV testing strategies such as voluntary counselling and testing (VCT), provider-initiated testing and counselling (PITC), diagnostic testing and counselling, home-based counselling and testing, and self-testing are available in SSA countries, HIV testing programs do not specifically target older adults [16, 21, 22]. Also, studies have uncovered that older PLWH in SSA are often unaware of treatment options and availability and they have inadequate access to health services [15]. However, most of these studies were conducted in Eastern and Southern Africa and do not report new HIV infections. In Togo, as in most SSA countries, national adult population surveys on HIV are generally limited to the 15–49 age group and in 2018, the prevalence of HIV was 2.3% [23]. According to the UNAIDS, it was estimated that among the 120,000 PLWH in Togo in 2019, 20,000 (16.7%) were aged ≥50 years [2]. To our knowledge, only one study has focused on the burden of the HIV epidemic among older adults in Togo and reported extrapolated prevalence based on existing data in 2010 [24]. The present study aimed to describe uptake of HIV testing, as well as HIV prevalence and its associated factors among older adults aged ≥50 years in health facilities in Togo.

## Methods

### Study design and settings

This cross-sectional study was conducted in Togo, a country of West Africa which covers an area of 56,800 Km$^2$ with an average density of 145 inhabitants per square kilometer. The population was 7.89 million in 2018, of which 50.2% are women. Most of the population is young (60% of Togolese are under 25 years of age), and lives in rural areas (62%). Togo health system has a three-level pyramid structure: central, intermediate and peripheral levels. For each level, there are administrative and healthcare delivery components. Regarding health care for older adults, no geriatric wards are available in the country.

This study was part of the Study of Hospitalized Older adults in Togo (SHOT) which aimed to describe comorbidities and mortality after hospitalization among older adults aged ≥50 years in Togo [25]. The SHOT project was carried out in two steps: face-to-face interview at baseline during hospitalization and telephone interviews at one (M1), three months (M3), six months (M6), and twelve months (M12) after hospital discharge. Baseline interviews were conducted from February 2018 to June 2019 in the medicine and surgery hospitalization wards of the three main tertiary level hospitals and secondary level hospitals according to the health pyramid in Togo. Results of HIV testing which was offered at baseline during hospitalization are reported in the present cross-sectional study.

### Study population and sample size

Patients aged≥50 years who were hospitalized in adult medical and surgical wards of participating hospitals during the study period were eligible to participate in the study. Exclusion criteria were inability to provide consent for HIV testing due to altered mental status or critical medical condition.

Since no data on HIV testing uptake among older adults were available in Togo, the sample size calculation was based on an expected HIV testing uptake in older adults aged ≥50 years of 73% based on the estimate reported in health facilities in SSA [26], with a precision of 4%, a significance level set at 5%, and a non-response rate of 10%; the minimum sample size was estimated at 522 patients. This sample size was also adequate for the estimation of HIV prevalence among older adults based on the following assumptions: an expected HIV prevalence in older adults aged ≥50 years of 2.3% based on the estimate reported in the general population (15–49 years) in Togo [23], with a precision of 2%, a significance level set at 5%, and a non-response rate of 10%, yielding a minimum sample size of 237 patients.

### Data collection

Prior to the study, the ten final-year medical students who enrolled the patients received training on HIV testing issues (counselling, obtaining consent, delivering post-test results in coordination with on-site HIV management staff) and on the interview questionnaire. All eligible patients were informed of the study procedures and were invited to participate. After informed consent, sociodemographic data and information regarding HIV testing history, and knowledge of their principal sexual partner's HIV status were recorded. For participants who reported being HIV positive or being tested during their stay at the hospital, medical records were assessed to collect information on their HIV testing history and status.

### HIV testing

HIV testing was performed according to the national algorithm in Togo, where HIV testing strategies are based on VCT and PICT. PICT is proposed to pregnant women, children aged

18–59 months, key populations and to all other adults. For the present study, HIV testing was proposed to all patients with and without history of HIV testing. Study participants who refused to be surveyed were asked to provide the reasons for their refusal.

A small amount of blood was collected by finger prick to perform rapid point-of-care HIV testing using Alere Determine[TM] HIV1/2 Ag/Ab Combo (Alere Medical Co. Ltd., Chiba, Japan). Each HIV positive test was confirmed with another HIV rapid test, the First Response® HIV 1-2-O Card Test (Premier Medical Corporation Pvt. Ltd., Maharashtra, India). In case of discordant results, venous blood samples were drawn and tested with the INNO-LIA® HIV I/II Score (20T) (Fujirebio, Göteborg, Sweden) line immunoassay at the main HIV laboratory research unit, the Molecular Biology Laboratory (BIOLIM) at the University of Lomé.

Counselling and risk prevention interventions were provided to patients who were tested negative for HIV. Positive results were disclosed by a senior medical officer in charge in the service where newly HIV positive patients were diagnosed. Senior medical officers also emphasized the benefits of medical care and treatment before referring newly diagnosed HIV patients to the HIV care center.

## Statistical analysis

Descriptive statistics were performed and results were presented with frequency tabulations and percentages for categorical variables. Quantitative variables were presented as medians with their interquartile range (IQR). HIV prevalence was estimated with corresponding 95% confidence interval (95%CI) based on estimation of confidence interval for a proportion [27]. Comparisons of medians were performed using Mood's Median test as it is more resistant to outliers and is more appropriate when distributions in the sub-populations being compared differ (one symmetrical and the other asymmetrical). Proportions were compared using Chi-squared test when sample size of all cells was greater than 5 or Fisher's exact test when at least one cell was below 5 [28]. Binary logistic regression analyses were performed to identify factors associated with HIV infection among older adults. In the univariable logistic regression, variables with a $p$-value $< 0.20$ were fitted into the multivariable analyses. A backward procedure approach was performed for selection of variables and adjusted odds ratio (aOR) were reported with their 95% CI. All analyses were performed using R® software. The significance level was set at 5%.

## Ethical considerations

All eligible patients received detailed information about the study purpose and procedures, voluntary participation, HIV testing, potential risks and protections. Recruited patients provided written signed informed consent prior to the administration of the questionnaire and additional consent was obtained prior to HIV testing. This study was approved by the "Comité de Bioéthique pour la Recherche en Santé (CBRS)" (Bioethics Committee for Health Research) from the Togo Ministry of Health (n˚09/2018/CBRS).

## Results

A total of 659 eligible older adults were recruited for the purpose of the SHOT project during the study period. Forty older adults were unable to provide consent for HIV testing because of their general health status (coma, stroke) and a total of 619 adults were offered HIV testing and included in the final analysis.

## Socio-demographic characteristics

The median age of the 619 participants was 61 years, (IQR: 55–70) (Table 1). Women represented 43.9% of the sample and they were more likely to be widowed (p<0.001) and have no formal or primary level education (p<0.001). Almost three quarters (73.8%) of the study population did not have health insurance. Men were more likely to have health insurance (p<0.001) and have higher monthly income (p<0.001) compared with women.

**Table 1. Socio-demographic characteristics of hospitalized older adults in Togo in 2018–2019 (N = 619).**

| | Total (N = 619) | | Male (N = 347) | | Female (N = 272) | | P-value |
|---|---|---|---|---|---|---|---|
| | N | % | N | % | N | % | |
| **Age (years), median (IQR)** | 61 (55–70) | | 60 (55–68) | | 62 (57–70) | | 0.002* |
| **Age (years)** | | | | | | | 0.021** |
| 50–59 | 262 | 42.3 | 161 | 46.4 | 101 | 37.1 | |
| ≥ 60 | 357 | 57.7 | 186 | 53.6 | 171 | 62.9 | |
| **City of residence** | | | | | | | 0.513** |
| Lomé | 359 | 58.0 | 204 | 58.8 | 155 | 57.0 | |
| Other cities | 182 | 29.4 | 104 | 30.0 | 78 | 28.7 | |
| MD | 78 | 12.6 | 36 | 11.2 | 39 | 14.3 | |
| **Marital status** | | | | | | | <0.001*** |
| Married/Living with a partner | 403 | 65.1 | 304 | 87.6 | 99 | 36.4 | |
| Widow | 170 | 27.5 | 27 | 7.8 | 143 | 52.6 | |
| Living alone | 43 | 6.9 | 14 | 4.0 | 29 | 10.7 | |
| MD | 3 | 0.5 | 2 | 0.6 | 1 | 0.3 | |
| **Education level** | | | | | | | <0.001** |
| No formal education | 173 | 27.9 | 48 | 13.8 | 125 | 46.0 | |
| Primary school | 158 | 25.5 | 88 | 25.4 | 70 | 25.7 | |
| Secondary/ University | 275 | 44.4 | 204 | 58.8 | 71 | 26.1 | |
| MD | 13 | 2.2 | 7 | 2.0 | 6 | 2.2 | |
| **Main source of monthly revenue** | | | | | | | <0.001** |
| None | 31 | 5.0 | 11 | 3.2 | 20 | 7.4 | |
| cFinancial aid | 184 | 29.7 | 71 | 20.5 | 113 | 41.5 | |
| Pension fund | 103 | 16.6 | 83 | 23.9 | 20 | 7.4 | |
| Income generating activity | 278 | 45.0 | 169 | 48.7 | 109 | 40.1 | |
| MD | 23 | 3.7 | 13 | 3.7 | 10 | 3.6 | |
| **Monthly income (Euros)** | | | | | | | <0.001** |
| < 53 | 136 | 22.0 | 58 | 16.7 | 78 | 28.7 | |
| 53–152 | 329 | 53.2 | 189 | 54.5 | 140 | 51.5 | |
| ≥ 153 | 118 | 19.1 | 84 | 24.2 | 34 | 12.5 | |
| MD | 36 | 5.7 | 16 | 4.6 | 20 | 7.3 | |
| **Health insurance** | | | | | | | <0.001*** |
| Yes | 157 | 25.4 | 120 | 34.6 | 37 | 13.6 | |
| No | 457 | 73.8 | 225 | 64.8 | 232 | 85.3 | |
| MD | 5 | 0.8 | 2 | 0.6 | 3 | 1.1 | |

*Mood's median test

**Chi-square test

***Fisher's exact test; IQR: Interquartile range; MD: missing data.

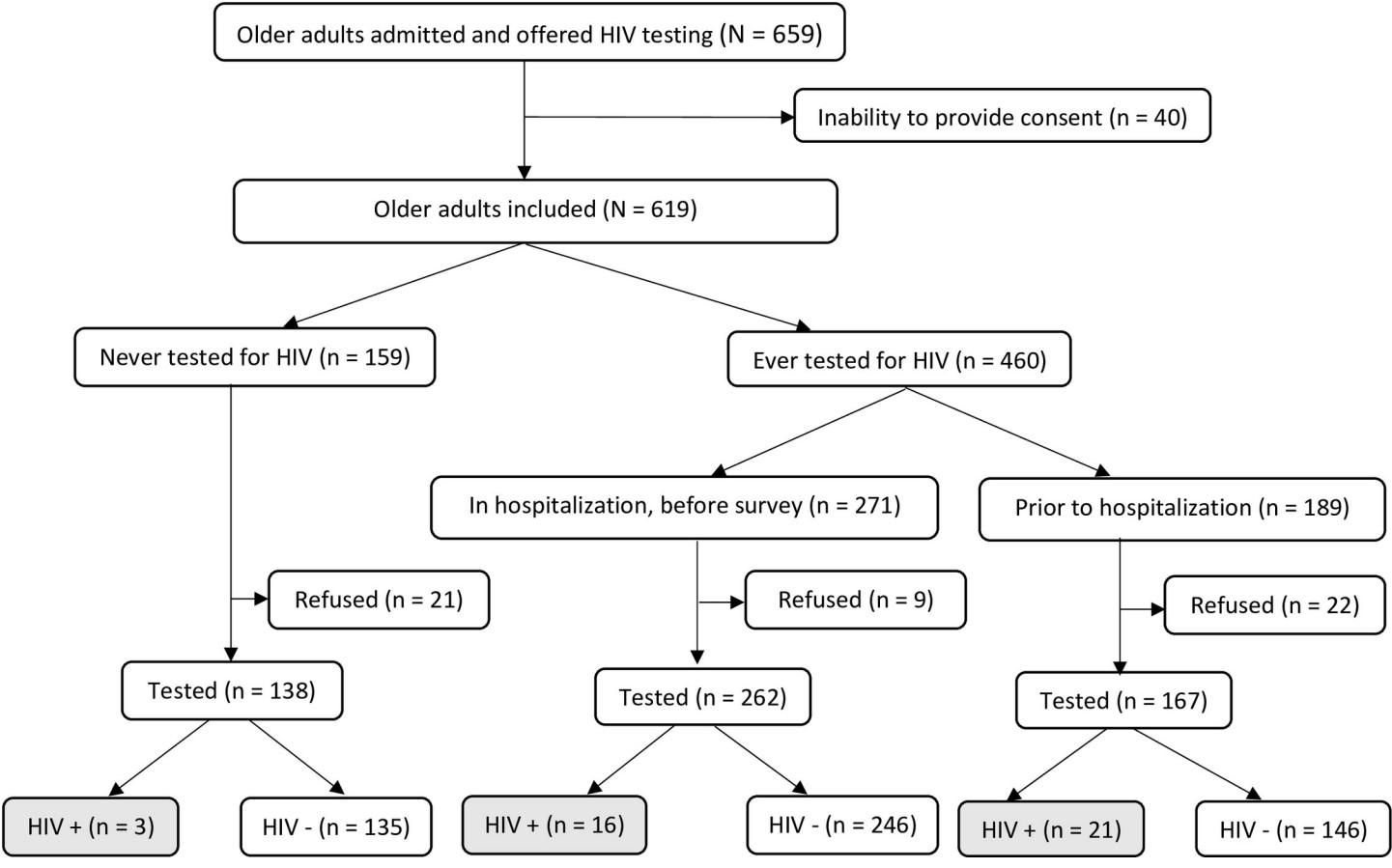

**Fig 1. HIV testing history and uptake among hospitalized older adults in Togo in 2018–2019.**

## History of HIV testing and uptake

Among the 619 older adults who were included in the study, 159 (25.7%) had never been tested for HIV and only 271 (43.7%) were tested during hospitalization, prior to our survey. In total, 91.6% (567/619) accepted HIV testing while 8.4% (52/619) refused to be tested; there was a statistical difference of HIV refusal proportions (p = 0.018) according to HIV history. Main reasons for HIV test refusal were 'I do not want to be tested again', 'I am not at risk', or there was no specific reason. History of HIV testing and uptake are summarized in Fig 1.

## HIV prevalence

Forty hospitalized older adults (of 567 tested) were positive for HIV, yielding a prevalence of 7.1%, 95%CI: [5.1–9.5]. Among HIV seropositive older adults, 23 (57.5%) were newly diagnosed as a result of this testing.

HIV prevalence was higher among older adults who were living alone (22.2%) compared with those who were widowed (5.7%) or married/living with a partner (5.9%) (p<0.001). Also, HIV prevalence was higher in those aged 50–59 years compared with their older counterparts (11.0% and 4.2%; p = 0.002). There was no statistical difference according to sex (6.9% in males, 7.2% in females; p = 0.954).

**Table 2. Factors associated with HIV prevalence among hospitalized older adults in Togo in 2018–2019.**

| | HIV prevalence (%) N = 567 | Univariable analysis | | | Multivariable (initial model) analysis | | | Multivariable (final model) analysis | | |
|---|---|---|---|---|---|---|---|---|---|---|
| | | OR | 95%CI | p-value | aOR | 95%CI | P-value | aOR | 95%CI | p-value |
| **Age (years)** | | | | | | | | | | |
| ≥ 60 | 4.2 | 1 | | | 1 | | | 1 | | |
| 50–59 | 11.0 | 3.12 | [1.58–6.25] | 0.001 | 3.33 | [1.58–7.14] | 0.002 | 3.12 | [1.51–6.66] | 0.002 |
| **Sex** | | | | | | | | | | |
| Male | 6.9 | 1 | | | 1 | | | 1 | | |
| Female | 7.2 | 1.07 | [0.56–2.05] | 0.832 | 0.65 | [0.27–1.50] | 0.318 | 0.72 | [0.32–1.59] | 0.418 |
| **Wards** | | | | | | | | | | |
| Medicine | 8.0 | 1 | | | 1 | | | 1 | | |
| Surgery | 3.7 | 0.39 | [0.11–1.0] | 0.079 | 0.37 | [0.11–0.99] | 0.074 | 0.38 | [0.11–0.99] | 0.075 |
| **Education level** | | | | | | | | | | |
| No formal education | 7.3 | 1 | | | 1 | | | - | | |
| Primary school | 6.4 | 0.84 | [0.34–2.05] | 0.710 | 0.66 | [0.25–1.70] | 0.391 | - | - | - |
| Secondary/University | 7.6 | 1.03 | [0.49–2.24] | 0.944 | 0.71 | [0.30–1.73] | 0.442 | - | - | - |
| **Marital status** | | | | | | | | | | |
| Married/Living with a partner | 5.9 | 1 | | | 1 | | | 1 | | |
| Widowed | 5.7 | 1.23 | [0.54–2.65] | 0.604 | 1.75 | [0.66–4.52] | 0.252 | 1.79 | [0.68–4.60] | 0.231 |
| Living alone | 22.2 | 6.23 | [2.61–14.21] | <0.001 | 5.92 | [2.29–14.87] | <0.001 | 5.83 | [2.26–14.53] | <0.001 |

95% CI: 95% confidence interval; OR: Odds Radio; aOR: adjusted Odds Ratio.

In multivariable analysis, two factors were associated with HIV infection: marital status (living alone) (aOR = 5.83; 95%CI = [2.26–14.53]) and being <60 years of age (aOR = 3.12; 95% CI = [1.51–6.66]). These results are presented in the Table 2.

## Discussion

The present study explored uptake of HIV testing and the prevalence of HIV infection among hospitalized older adults in tertiary and secondary hospitals in Togo in 2018–2019. More than a quarter of hospitalized older adults had never previously tested for HIV. Among participants who accepted HIV testing, HIV prevalence was estimated at 7.1%, which was more than three times that of the general population (2.3% in 2018) among people aged 15–49 years. Factors associated with positive HIV status were living alone and being aged <60 years old.

Reaching the first 90–90% of people living with HIV know their status–remains a challenge in SSA, especially for older adults. In rural Uganda, proportion of lifetime HIV testing among older adults ≥50 years was 82% and recent (last 12 months) HIV testing was 53% [16], while in rural area in Tanzania, lower HIV testing rate (ever tested) of 11.4% was reported in a cross-sectional study which used baseline data of the Ifakara MZIMA cohort study in 2012/13 [18]. Experience of HIV testing in older adults may depend on where they tested (hospital or community setting) or whether they actively sought the testing or not [22]. In Botswana, among the barriers to HIV testing, 76.8% older adults felt that they were not at risk of HIV infection or did not have time to go to the testing center (6.5%) [29]. Also, older adults may mistake HIV symptoms for those of normal ageing and not consider HIV as a potential cause [30].

Prior to this survey conducted in selected hospital settings, only two older adults in five (43.7%) were offered and tested for HIV during their current hospitalization. Although sexual transmission has been reported as the main route of transmission among older adults [6], they

are often perceived as asexual patients by health care providers and are less likely to be offered HIV testing [31, 32]. In Kenya, older adults expressed concerns with ageist discrimination specifically in hospital settings, characterized by providers' reluctance or refusal to test [22]. Besides, health care providers may feel uncomfortable discussing sexual history, especially with older adults [33]. HIV testing and prevention initiatives have historically focused on younger population and key populations. PITC should also include older adults irrespective of their marital status to detect undiagnosed cases at an early stage. Also, an alternative approach with HIV self-testing could be proposed [34].

Some studies have reported data on HIV prevalence in Africa among older adults aged ≥50 years. Extrapolation from existing data from Demographic and Health Surveys (DHS) in SSA yielded a prevalence of 4.0% among people aged >50 years [24]. Surveys that have included older adults have reported HIV prevalence ranging from 5% among 50–64 years old in Kenya to 13% among 50–54 years old in South Africa [35]. In Swaziland, prevalence was 6.4% among adults aged 50 years or older, and 13% among men, and 7% among women aged 60–64 years [35]. While comparing with HIV prevalence observed in the general population, another review of data from 40 DHS conducted in 27 SSA countries between 2003 and 2012 found that HIV prevalence among adults (45–59 years) was higher than in the general adult population, except in Democratic Republic of Congo, Ethiopia, Mozambique, Sierra Leone, and Swaziland [36]. In our study, one hospitalized older adult in 14 (7.1%) was infected with HIV and similarly, a study conducted in a hospital in Dares-Salam in Tanzania reported high prevalence of 15% among those aged 55 years and older [37]. Thus, HIV prevalence among older adults varies greatly across countries in SSA. However, comparisons should be made with caution since the recruitment settings or strategies differ from one country to another. Nevertheless, these results showed the need to focus prevention strategy on older adults.

Age and marital status were associated with HIV infection in the present study. Few studies have reported data on HIV prevalence stratified by age among older adults in SSA. In South Africa, a secondary data analysis based on the 2012 population-based nationally representative multi-stage stratified cluster random household sample showed that increased risk of HIV was significantly associated with age group 25–49 years and those 50 years and older compared with young males 15–25 years [38]. In our study, the odds of being infected with HIV was three times higher among older adults aged 50–59 years than in their counterparts aged 60 years and over. Globally, HIV prevalence appears to peak among people aged 45–60 years in SSA. In Zimbabwe, HIV prevalence was high among men (23.4%) and women (21.3%) aged 45–54 years compared to those aged 15–44 years with 11.0% and 17.3%, respectively [14]. Based on available data in South Africa, the prevalence of HIV was 10.4% in males aged 50–54 years, 6.2% in males aged 55–59 years, and 3.5% in males aged ≥60 years [24]. Similar trends were reported in Kenya, Rwanda, and in Uganda [24]. Intergenerational sex (having sex with partners at least 10 years younger) without condom is a risky and frequent sexual behavior among older adults [13, 39], which could explain this high prevalence. Also, this increased prevalence may be related to practices such as wife inheritance, which are common in SSA and where a widow marries a deceased's relative [13].

Older adults who were living alone had six times the odds of being infected with HIV compared to their counterparts who were married or living with a partner. Similarly in South Africa, it was reported that married individuals who were living with their spouse had significantly reduced odds of being HIV positive compared to all other marital status groups [40]. This could be explained by the fact that single individuals are more likely to have high numbers of sexual partners and that older adults inconsistently use condom during sexual intercourse [15, 41].

This study has some limitations. First, we used the national algorithm for HIV diagnosis, which does not allow to identify recent HIV infection [42]. Second, we recruited older adults in clinical settings; therefore, HIV prevalence estimates could not be generalized to all older adults in Togo and could be overestimated. However, this study documents missed opportunities of HIV testing among hospitalized older adults and the results showed the need to focus prevention strategy in older adults. Also, it provided an overview of HIV prevalence in hospitals in Togo and the need to test all older adults.

## Conclusion and recommendations

This study explored HIV testing uptake, HIV prevalence and its associated factors among older adults in hospitals in Togo. We observed high prevalence of HIV. Achieving the 90:90:90 targets requires identifying the largest possible number of HIV-positive individuals, including older adults. In addition, management of HIV infection among older adults deserves great attention as for non-communicable diseases. Thus, there is an urgent need to integrate older adults into responses to the HIV epidemic and to strengthen targeted prevention care and treatment in this population. Futures studies should better document virological responses in older adults in Africa since immunological responses seems to be influenced by older age [43]. Moreover, the relationship between HIV infection and sexual practices among older adults should be better documented especially in West Africa.

## Supporting information

**S1 Dataset. HIV testing uptake and prevalence among hospitalized older adults in Togo: A cross-sectional study.**
(XLSX)

## Acknowledgments

We are thankful to the older adults who accepted to participate in this study. We are also thankful to the medical students of the 'Faculté des Sciences de la Santé-Université de Lomé' who performed data collection for the study.

We are deeply indebted to the "Centre Africain de Recherche en Epidémiologie et en Santé Publique" (CARESP) and the "Programme National de Lutte contre le sida, les hépatites virales et les infections sexuellement transmissibles du Togo" (PNLS/HV/IST) who provided technical and logistical support to this work.

## Author Contributions

**Conceptualization:** Fifonsi Adjidossi Gbeasor-Komlanvi, Didier Koumavi Ekouevi.

**Data curation:** Fifonsi Adjidossi Gbeasor-Komlanvi, Martin Kouame Tchankoni.

**Formal analysis:** Fifonsi Adjidossi Gbeasor-Komlanvi, Wendpouire Ida Carine Zida-Compaore, Arnold Junior Sadio, Martin Kouame Tchankoni, Didier Koumavi Ekouevi.

**Investigation:** Fifonsi Adjidossi Gbeasor-Komlanvi, Balakiyem Magnim Kadangha, Mounerou Salou.

**Methodology:** Fifonsi Adjidossi Gbeasor-Komlanvi, Mounerou Salou, Anoumou Claver Dagnra, Didier Koumavi Ekouevi.

**Project administration:** Fifonsi Adjidossi Gbeasor-Komlanvi, Wendpouire Ida Carine Zida-Compaore, Arnold Junior Sadio, Balakiyem Magnim Kadangha, Mounerou Salou, Anoumou Claver Dagnra, Didier Koumavi Ekouevi.

**Resources:** Anoumou Claver Dagnra, Didier Koumavi Ekouevi.

**Software:** Martin Kouame Tchankoni.

**Supervision:** Fifonsi Adjidossi Gbeasor-Komlanvi, Wendpouire Ida Carine Zida-Compaore, Arnold Junior Sadio, Balakiyem Magnim Kadangha, Didier Koumavi Ekouevi.

**Validation:** Fifonsi Adjidossi Gbeasor-Komlanvi, Mounerou Salou, Anoumou Claver Dagnra, Didier Koumavi Ekouevi.

**Visualization:** Wendpouire Ida Carine Zida-Compaore, Arnold Junior Sadio, Martin Kouame Tchankoni, Didier Koumavi Ekouevi.

**Writing – original draft:** Fifonsi Adjidossi Gbeasor-Komlanvi, Wendpouire Ida Carine Zida-Compaore, Arnold Junior Sadio, Martin Kouame Tchankoni, Didier Koumavi Ekouevi.

**Writing – review & editing:** Balakiyem Magnim Kadangha, Mounerou Salou, Anoumou Claver Dagnra.

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
