## [Decision Letter · Decision Letter 0]

20 Jul 2020

PONE-D-20-09735

HIV testing uptake and prevalence among hospitalized older adults in Togo: a cross-sectional study

PLOS ONE

Dear Dr. Gbeasor-Komlanvi,

Thank you for submitting your manuscript to PLOS ONE. We wish to apologize for the delayed review of this manuscript; reviewer availability has been restricted recently and a number of reviewers were unable to complete their review after initially agreeing, which delayed the review process.

After careful consideration, we feel that it has merit but does not fully meet PLOS ONE’s publication criteria as it currently stands. Therefore, we invite you to submit a revised version of the manuscript that addresses the points raised during the review process.

The reviewers highlighted some details of the study design/recruitment which require clarification, additional literature that should be reviewed and cited and elements of grammar and language that will require attention before the manuscript can be accepted.

We look forward to receiving your revised manuscript.

Kind regards,

Anna C Hearps

Academic Editor

PLOS ONE

Journal Requirements:

"This work was supported by the “Centre Africain de Recherche en Epidémiologie et en Santé

Publique” (CARESP) and the “Programme National de Lutte contre le sida, les hépatites virales et les infections sexuellement transmissibles du Togo” (PNLS/HV/IST)."

Reviewers' comments:

Reviewer's Responses to Questions

**Comments to the Author**

1. Is the manuscript technically sound, and do the data support the conclusions?

Reviewer #1: Partly

Reviewer #2: Yes

2. Has the statistical analysis been performed appropriately and rigorously? 

Reviewer #1: Yes

Reviewer #2: Yes

3. Have the authors made all data underlying the findings in their manuscript fully available?

Reviewer #1: Yes

Reviewer #2: Yes

4. Is the manuscript presented in an intelligible fashion and written in standard English?

Reviewer #1: No

Reviewer #2: Partly

5. Review Comments to the Author

Reviewer #1: HIV testing uptake and prevalence among hospitalized older adults in Togo: a cross-sectional study

This is an interesting paper with surprising results from a region with low HIV prevalence. The paper continues to add to the body of knowledge on HIV among the older adults. Although the paper provides useful information on HIV testing uptake and prevalence among hospitalized older adults, the manuscript requires major improvement before its consideration for publication.

Title: The title is well written providing information on the population and the study setting as well as the design employed. Since this study was not carried out in the whole of Togo, mentioning Lomé, Togo in the title would be important.

Abstract: The abstract is well structured and concise. The results section needs a better clarity, who were the 340 that were offered the testing. I have read through the text and looked at Figure 1 to identify this number, but I am unable to. Of those offered testing, 84.7% took the test (figure calculated based on the decline %), would this be considered low uptake? Compared to what?

Introduction: Generally, the introduction section is not focused to what the manuscript is communicating. Would have liked to read more on the importance of testing especially among the older adults, the strategies employed to test older people. There is literature available in sub-Saharan Africa on prevalence of HIV among older adults. I provide a few articles below < 4 years. The introduction section would also benefit from copy-editing services and updating the recent global data on HIV and aging (2019 http://aidsinfo.unaids.org/ also provide estimated data for the number of 50+ living with HIV in Togo, at about 14,000).

• Wandera SO, Kwagala B, Maniragaba F. Prevalence and determinants of recent HIV testing among older persons in rural Uganda: a cross-sectional study. BMC Public Health. 2020 Dec 1;20(1):144.

• Swai SJ, Damian DJ, Urassa S, Temba B, Mahande MJ, Philemon RN, Msuya SE. Prevalence and risk factors for HIV among people aged 50 years and older in Rombo district, Northern Tanzania. Tanzania Journal of Health Research. 2017;19(2).

• Mtowa A, Gerritsen AA, Mtenga S, Mwangome M, Geubbels E. Socio-demographic inequalities in HIV testing behaviour and HIV prevalence among older adults in rural Tanzania, 2013. AIDS care. 2017 Sep 2;29(9):1162-8.

• Matlho K, Randell M, Lebelonyane R, Kefas J, Driscoll T, Negin J. HIV prevalence and related behaviours of older people in Botswana—secondary analysis of the Botswana AIDS Impact Survey (BAIS) IV. African Journal of AIDS Research. 2019 Jan 2;18(1):18-26.

• Rosenberg MS, Gómez-Olivé FX, Rohr JK, Houle BC, Kabudula CW, Wagner RG, Salomon JA, Kahn K, Berkman LF, Tollman SM, Bärnighausen T. Sexual behaviors and HIV status: a population-based study among older adults in rural South Africa. Journal of acquired immune deficiency syndromes (1999). 2017 Jan 1;74(1):e9.

Methods: The study design for the main study (SHOT) is not clearly stated to provide an understanding of how the current study is embedded in it. It is unclear if the testing done in this cross-sectional study was separate from the SHOT one. A description of Togo’s and the healthcare facility’s HIV testing protocols would be relevant in understanding the study setting.

I do not understand the relevance of sample size calculation if the population recruited in SHOT study was also recruited in this current study. Also, the mention of the lack of prevalence data (line 126) negates what the authors mention in line 101-103.

It is unclear what exactly was done in this study. Was testing offered for this specific study or for SHOT? If for this cross-sectional study, you mention (line 142-144) that the test results were abstracted from the medical records. Who then did you consider having consented for the test, and who were considered to have declined (15.3%) to take the test? The tests that were done at BIOLIM, how long did those take to provide the results. I assume the other results were provided at the point of care.

Which statistical analysis software was used to conduct the analysis?

Results: Were there reasons for inability to provide consent by the 40 participants? It would be important to state this as it may have ‘potential’ bias to your study.

Table 1: What does ‘living alone’ mean, especially for an older adult. Does this mean they were never married in their lives, or had a partner? Are these what are considered divorced/separated?

Section on History of HIV testing and uptake is a confusing especially if the separation between what was done for this study and that of SHOT is not made. Of interest is the 9 who refused to be retested, based on the medical records abstracted, were they positive or negative? The 15.3% that declined to test, could the results be limited by self-selection of testing based on HIV risk history? So, what was the HIV testing uptake for this study? Tenses need to be corrected in the paragraph.

Section on HIV prevalence, I do not understand the sentence (line 211-212). I am still trying to figure out who were the 340 that were offered testing? The sentence, “of the 40 older adults who tested positive for HIV, ….” Who tested them? Or did you mean “of the 40 older adults infected with HIV”? I ask this because it is not clear whether those already known positives were tested again?

Unless it is clarified who ‘living alone’ meant, the results from this study should be cautiously interpreted.

Discussion: This section requires rewriting to focus on the results of the study. Interpreting the study results in the context of the available literature is key. References are largely made to studies in the US and other HIC yet there are relevant data for comparison in LMIC and I would encourage the authors to read more.

• Ahmed S, Bärnighausen T, Daniels N, Marlink R, Roberts MJ. How providers influence the implementation of provider-initiated HIV testing and counseling in Botswana: a qualitative study. Implement Sci. 2015;11(1):18

• Kiplagat J, Huschke S. HIV testing and counselling experiences: a qualitative study of older adults living with HIV in western Kenya. BMC geriatrics. 2018 Dec;18(1):1-0.

• Wachira J, Ndege S, Koech J, Vreeman RC, Ayuo P, Braitstein P. HIV testing uptake and prevalence among adolescents and adults in a large home-based HIV testing program in Western Kenya. JAIDS Journal of Acquired Immune Deficiency Syndromes. 2014 Feb 1;65(2):e58-66.

Conclusion: The statement on stigma (line 319-320) is not supported by the study results. Sentence (line 321-323) is also not supported by the study findings. No information related to HIV response is provided that informs the lack of or inadequate integration of older adults. The last sentence (323-326), authors need to read more on immunologic response, sexual behaviors of older adults in SSA as literature is available.

Reviewer #2:

This is an interesting article describing the HIV prevalence in older adults in a hospitalized setting in Togo, rates of HIV testing and barriers to regular testing in this population. Aside from some minor grammatical errors (see below) it was a well written and constructed article, and highlighted an important population requiring attention in our international efforts to obtain the 90-90-90 goals.

Comments:

It is sometimes unclear what denominator is used for the calculation of percentage results, and this is occasionally inconsistent. For example, paragraph beginning Line 210, the overall HIV prevalence is stated as 7.0%, but later the prevalence for males and females is stated as 5.7% and 6.1% (both values lower than 7.0%. This should be corrected and clarified.

If a statement is made that something was higher in a certain subgroup, then then the comparator group should be stated (eg Line187-9: “Women were more likely to be widowed…” assume as compared to men but should be stated. Also line 214)

Some minor typographical and grammatical changes required:

Line 73: Rather than “large” perhaps say “widespread”  or “large-scale” access to ART

Line 75: Change longer to long

Line 77: Change HIV new to new HIV

Line 86: Change to: from a low rate

Line 98” Remove mainly

Line 136: Remove a from the end of line

Line 170: Add s to variable

Table 1: Maybe include median (IQR) age of male vs female participants?

Paragraph beginning Line 200:  Use past tense so change have to had throughout

Line 251: Change to initiatives (plural)

Line 272: Change to An alternative approach…

Line 309: Change recruit to recruited (past tense)

Line 311: Remove “first”

6. PLOS authors have the option to publish the peer review history of their article (what does this mean?). If published, this will include your full peer review and any attached files.

Reviewer #1: No

Reviewer #2: No

---

## [Author Response · Author response to Decision Letter 0]

18 Sep 2020

Responses to reviewers

Editors comments

1) Thank you for stating the following in the Funding Section of your manuscript:

"This work was supported by the “Centre Africain de Recherche en Epidémiologie et en Santé

Publique” (CARESP) and the “Programme National de Lutte contre le sida, les hépatites virales et les infections sexuellement transmissibles du Togo” (PNLS/HV/IST)."

Authors: We thank the Editors for this remark. We removed the funding-related text from the manuscript. We rather thank the CARESP and PNLS/HV/IST for their support.

2) Please include captions for your Supporting Information files at the end of your manuscript, and update any in-text citations to match accordingly. Please see our Supporting Information guidelines for more information: http://journals.plos.org/plosone/s/supporting-information.

Authors: We included captions for the Supporting Information (dataset) at the end of the manuscript. 

Reviewers' comments

Reviewer #1

HIV testing uptake and prevalence among hospitalized older adults in Togo: a cross-sectional study

This is an interesting paper with surprising results from a region with low HIV prevalence. The paper continues to add to the body of knowledge on HIV among the older adults. Although the paper provides useful information on HIV testing uptake and prevalence among hospitalized older adults, the manuscript requires major improvement before its consideration for publication.

Authors: We thank the reviewer for this comment for this encouraging comment. 

1) Title: The title is well written providing information on the population and the study setting as well as the design employed. Since this study was not carried out in the whole of Togo, mentioning Lomé, Togo in the title would be important.

Authors: The study was carried out in hospitals of the three main tertiary hospitals in Togo and hospitals from secondary levels which are located in the health region of Lomé and other health regions in Togo. Health structures of the first level of the health pyramid in Togo do not perform hospitalization. We completed this information in the methods section of the abstract in the revised version of the manuscript. 

2) Abstract: The abstract is well structured and concise. The results section needs a better clarity, who were the 340 that were offered the testing. I have read through the text and looked at Figure 1 to identify this number, but I am unable to. 

Authors: We thank the reviewer for this comment. These results have been clarified in the revised manuscript. A total of 567 older adults were offered HIV testing. 

3) Of those offered testing, 84.7% took the test (figure calculated based on the decline %), would this be considered low uptake? Compared to what?

Authors: We agree with the reviewer. This sentence has been reformulated in the revised manuscript.

4) Introduction: Generally, the introduction section is not focused to what the manuscript is communicating. Would have liked to read more on the importance of testing especially among the older adults, the strategies employed to test older people. There is literature available in sub-Saharan Africa on prevalence of HIV among older adults. I provide a few articles below < 4 years. The introduction section would also benefit from copy-editing services and updating the recent global data on HIV and aging (2019 http://aidsinfo.unaids.org/ also provide estimated data for the number of 50+ living with HIV in Togo, at about 14,000).

• Wandera SO, Kwagala B, Maniragaba F. Prevalence and determinants of recent HIV testing among older persons in rural Uganda: a cross-sectional study. BMC Public Health. 2020 Dec 1;20(1):144.

• Swai SJ, Damian DJ, Urassa S, Temba B, Mahande MJ, Philemon RN, Msuya SE. Prevalence and risk factors for HIV among people aged 50 years and older in Rombo district, Northern Tanzania. Tanzania Journal of Health Research. 2017;19(2).

• Mtowa A, Gerritsen AA, Mtenga S, Mwangome M, Geubbels E. Socio-demographic inequalities in HIV testing behaviour and HIV prevalence among older adults in rural Tanzania, 2013. AIDS care. 2017 Sep 2;29(9):1162-8.

• Matlho K, Randell M, Lebelonyane R, Kefas J, Driscoll T, Negin J. HIV prevalence and related behaviours of older people in Botswana—secondary analysis of the Botswana AIDS Impact Survey (BAIS) IV. African Journal of AIDS Research. 2019 Jan 2;18(1):18-26.

• Rosenberg MS, Gómez-Olivé FX, Rohr JK, Houle BC, Kabudula CW, Wagner RG, Salomon JA, Kahn K, Berkman LF, Tollman SM, Bärnighausen T. Sexual behaviors and HIV status: a population-based study among older adults in rural South Africa. Journal of acquired immune deficiency syndromes (1999). 2017 Jan 1;74(1):e9.

Authors: We thank the reviewer for providing references on the topic. These comments have been taken into account and introduction section has been revised according to reviewer’s suggestions. 

5) Methods: The study design for the main study (SHOT) is not clearly stated to provide an understanding of how the current study is embedded in it. It is unclear if the testing done in this cross-sectional study was separate from the SHOT one. 

Authors: We provided more information about the study design. HIV testing was included in the SHOT study. We proposed HIV testing to all SHOT patients who could provide a consent. We reported results of the HIV testing in the present paper. 

6) A description of Togo’s and the healthcare facility’s HIV testing protocols would be relevant in understanding the study setting.

Authors: We thank the reviewer for this comment. We provided additional information on Togo’s and healthcare facility’s HIV protocols in the revised manuscript. 

7) I do not understand the relevance of sample size calculation if the population recruited in SHOT study was also recruited in this current study. Also, the mention of the lack of prevalence data (line 126) negates what the authors mention in line 101-103.

Authors: The sample size has been calculated to be sure that the number of recruited participants for SHOT may also produce conclusive results for the study on HIV. We also corrected the mention on the lack of the prevalence data. It was rather the lack of data on HIV testing uptake. 

8) It is unclear what exactly was done in this study. Was testing offered for this specific study or for SHOT? If for this cross-sectional study, you mention (line 142-144) that the test results were abstracted from the medical records. Who then did you consider having consented for the test, and who were considered to have declined (15.3%) to take the test?

Authors: HIV testing was offered to all eligible older adults. Medical records were assessed to collect information on HIV testing history and status. Thus, older adults who consented for the test were those who accepted to test, and those who refused to test were considered to have declined.

9) The tests that were done at BIOLIM, how long did those take to provide the results. I assume the other results were provided at the point of care. 

Authors: Point of care testing were performed to test for HIV. No discordant results were observed; thus, tests were not carried out at Biolim. 

10) Which statistical analysis software was used to conduct the analysis?

Authors: This information is provided in the methods section of the revised manuscript. 

11) Results: Were there reasons for inability to provide consent by the 40 participants? It would be important to state this as it may have ‘potential’ bias to your study.

Authors: We thank the reviewer for this comment. Details were provided in the methods section.

12) Table 1: What does ‘living alone’ mean, especially for an older adult. Does this mean they were never married in their lives, or had a partner? Are these what are considered divorced/separated?

Authors: The term ‘living alone’ means ‘no current partner’ and includes: never married, divorced/separated older adults.

13) Section on History of HIV testing and uptake is a confusing especially if the separation between what was done for this study and that of SHOT is not made. 

Authors: The section on history of HIV testing and uptake has been rewritten for a better understanding.

14) Of interest is the 9 who refused to be retested, based on the medical records abstracted, were they positive or negative? 

Authors: HIV testing results were not available in the medical records for the 9 older adults who refused to be retested. 

15) The 15.3% that declined to test, could the results be limited by self-selection of testing based on HIV risk history? So, what was the HIV testing uptake for this study? Tenses need to be corrected in the paragraph.

Authors: These comments have been taken into account in the revised version of the manuscript. The proportion of HIV testing refusal was recalculated and estimated at 8.4%. 

16) Section on HIV prevalence, I do not understand the sentence (line 211-212). I am still trying to figure out who were the 340 that were offered testing? 

Authors: We thank the reviewer for this comment. This section has been rewritten for a better understanding. 

17) The sentence, “of the 40 older adults who tested positive for HIV, ….” Who tested them? Or did you mean “of the 40 older adults infected with HIV”? I ask this because it is not clear whether those already known positives were tested again?

Authors: We retested known HIV positives older patients. We rephrased the sentence in the revised version of the manuscript. 

18) Unless it is clarified who ‘living alone’ meant, the results from this study should be cautiously interpreted.

Authors: This comment has been taken into account above and we provided a definition for the term “living alone”. 

19) Discussion: This section requires rewriting to focus on the results of the study. Interpreting the study results in the context of the available literature is key. References are largely made to studies in the US and other HIC yet there are relevant data for comparison in LMIC and I would encourage the authors to read more.

• Ahmed S, Bärnighausen T, Daniels N, Marlink R, Roberts MJ. How providers influence the implementation of provider-initiated HIV testing and counseling in Botswana: a qualitative study. Implement Sci. 2015;11(1):18

• Kiplagat J, Huschke S. HIV testing and counselling experiences: a qualitative study of older adults living with HIV in western Kenya. BMC geriatrics. 2018 Dec;18(1):1-0.

• Wachira J, Ndege S, Koech J, Vreeman RC, Ayuo P, Braitstein P. HIV testing uptake and prevalence among adolescents and adults in a large home-based HIV testing program in Western Kenya. JAIDS Journal of Acquired Immune Deficiency Syndromes. 2014 Feb 1;65(2):e58-66.

Authors: We thank the reviewer for providing the articles. Discussion has been rewritten taking into account Africa data sent by the reviewer. 

20) Conclusion: The statement on stigma (line 319-320) is not supported by the study results. Sentence (line 321-323) is also not supported by the study findings. No information related to HIV response is provided that informs the lack of or inadequate integration of older adults. The last sentence (323-326), authors need to read more on immunologic response, sexual behaviors of older adults in SSA as literature is available.

Authors: This comment has been taken into account and the conclusion section was rephrased. 

Achieving the 90:90:90 targets requires identifying the largest possible number of HIV-positive individuals, including older adults. Also, management of HIV infection among older adults deserves great attention especially with regards to non-communicable diseases. Thus, there is an urgent need to integrate older adults into responses to the HIV epidemic and to strengthen targeted prevention care and treatment in this population. Futures studies should better document virological responses in older people in Africa since immunological responses seems to be influenced by older age [40]. In addition, the relationship between HIV infection and sexual practices among older adults should be more documented especially in West Africa. 

Reviewer #2

This is an interesting article describing the HIV prevalence in older adults in a hospitalized setting in Togo, rates of HIV testing and barriers to regular testing in this population. Aside from some minor grammatical errors (see below) it was a well written and constructed article, and highlighted an important population requiring attention in our international efforts to obtain the 90-90-90 goals.

Authors: We thank the reviewer for this comment for this encouraging comment. 

1) It is sometimes unclear what denominator is used for the calculation of percentage results, and this is occasionally inconsistent. For example, paragraph beginning Line 210, the overall HIV prevalence is stated as 7.0%, but later the prevalence for males and females is stated as 5.7% and 6.1% (both values lower than 7.0%. This should be corrected and clarified.

Authors: We thank the reviewer for this comment. The denominator used for the calculation of HIV prevalence was N=567, excluding older adults who refused HIV testing. 

2) If a statement is made that something was higher in a certain subgroup, then then the comparator group should be stated (eg Line187-9: “Women were more likely to be widowed…” assume as compared to men but should be stated. Also line 214).

Authors: We thank the reviewer for this comment which has been taken into account in the revised version of the manuscript.

3) Table 1: Maybe include median (IQR) age of male vs female participants?

Authors: We provided median (IQR) age for male and female participants in Table 1 in the revised version of the manuscript.

4) Some minor typographical and grammatical changes required:

Line 73: Rather than “large” perhaps say “widespread” or “large-scale” access to ART

Line 75: Change longer to long

Line 77: Change HIV new to new HIV

Line 86: Change to: from a low rate

Line 98” Remove mainly

Line 136: Remove a from the end of line

Line 170: Add s to variable

Paragraph beginning Line 200: Use past tense so change have to had throughout

Line 251: Change to initiatives (plural)

Line 272: Change to An alternative approach…

Line 309: Change recruit to recruited (past tense)

Line 311: Remove “first”

Authors: We thank the reviewer for these corrections. A native English speaker proofread the revised version of the manuscript and grammatical errors were corrected.

---

## [Editor Report · Decision Letter 1]

16 Oct 2020

PONE-D-20-09735R1

HIV testing uptake and prevalence among hospitalized older adults in Togo: a cross-sectional study

PLOS ONE

Dear Dr. GBEASOR-KOMLANVI,

Thank you for submitting your revised manuscript to PLOS ONE. After careful consideration, we feel that it has merit but does not fully meet PLOS ONE’s publication criteria as it currently stands. Therefore, we invite you to submit a revised version of the manuscript that addresses the points raised during the review process.

Whilst we thank you for conscientiously addressing the Reviewer's suggestions on your manuscript and correcting questions of study design and accuracy, review of the revised version has unfortunately indicated further changes are required to render the article suitable for publication. Although these required changes are primarily textural, the review has indicated significant restructure of the Discussion is required which will be essential for the article to be accepted. Once these changes have been made, it would be useful to have the revised manuscript again reviewed by a native English speaker as you indicate you have done for the previous submission.

We look forward to receiving your revised manuscript.

Kind regards,

Anna C Hearps

Academic Editor

PLOS ONE

Additional Editor Comments (if provided):

The manuscript has been substantially improved with the changes made in response to the previous Reviewers' suggestions. The scientific validity of the data presented is sound, and the findings of interest. However, sections of the manuscript, particularly the Discussion, still require significant changes if the article is to be published. I have addressed some specific instances of phrasing and grammar that require attention as edits in the attached document, but also make some additional suggestions below to try and restructure the Discussion:

Lines 47-49: See text edits.

Line 49: 40/567 is 7.1% - please correct. Also, how is there a confidence interval for this result.

Line 53: The revised sentence should probably be reworded to eg “The majority of older adults in this study accepted testing for HIV and 4.1% of those tested were newly diagnosed with HIV as a result of this test” or similar.

Line 70: See text edit.

Line 72: This line is now repetitive of the sentences above. Please rephrase to state ART has contributed to the abovementioned increases in numbers of PLWH 50+.

Line 78: State the age definition of older adults in Reference 5.

Line 103: Here and elsewhere remove abbreviations where they are used only once.

Line 110-11: 20,000/120,000 does not equal 18.3%. Please check actual numbers and correct.

Line 115: States the study was carried out in Lome, but in response to reviewer 1 it is stated the study was carried out in both Lome and other health regions in Togo. Please clarify, and if only hospitals in Lome were involved amend the title as suggested by Reviewer 1.

Line 181: See comment above – it is not clear how a point prevalence has a 95% CI?

Line 206: p=<0.001 for both factors? Table 1 indicates other significant differences between men and women here eg monthly income, health insurance. Please mention these important differences in the text.

Line 211: Please define the Mood test in Methods.

Line 223-4: Perhaps flip this result and state that 23 (57.5%) were newly diagnosed as a result of this testing.

Table 1 and 2: Please correct age brackets to either 50-59 and ≥60 OR 50-60 and >60 (ie not 50-60 and ≥60).

Discussion:

The additional studies relevant to Africa which have been included in the revised Discussion are of interest to gain context of the data generated in the present study. However, as written this revised Discussion lacks structure and logical flow and is far too long. It would be of use to briefly mention the data pertaining to HIV prevalence/testing in older people in Africa and compare with the current findings, then discuss possible reasons and suggested interventions. Some specific suggestions include (but are not limited to):

Line 248: Remove sentence beginning “Data in…”

Line250-1: Where is the result indicating testing was not offered? State or remove sentence.

Line 252-4/Ref 27: If these data are not specific to older people, please remove.

Line 262: Remove sentence “refusal of testing…”

Line 264: Study in Zambia not relevant to testing in older people – remove.

Line 269-70: Can remove sentence.

Line 274-76: Rephrase to say HIV testing campaigns targeting older people may be required.

Lines 277-284: Could be removed.

Line 289: Can remove sentence beginning “The low testing…”

Line 294 (and also line 337): The statement that there is limited data on HIV prevalence in older adults is inconsistent with the studies cited after, and in the Introduction, which detail a number of studies. Please rephrase. This section (from lines 294 to 342) needs to be significantly shortened; group similar studies from different countries together, only include data relevant to older populations, critically compare and contrast results from this study with those previously reported and avoid long lists of prevalence data with minimal interpretation.

Lines 338 and 342 – these statements seem to contradict?

Line 362: The statement that HIV testing uptake was low doesn’t seem justified – remove.
---

## [Author Response · Author response to Decision Letter 1]

7 Dec 2020

Additional Editor Comments 

The manuscript has been substantially improved with the changes made in response to the previous Reviewers' suggestions. The scientific validity of the data presented is sound, and the findings of interest. However, sections of the manuscript, particularly the Discussion, still require significant changes if the article is to be published. I have addressed some specific instances of phrasing and grammar that require attention as edits in the attached document, but also make some additional suggestions below to try and restructure the Discussion.

Authors: We thank the editor for these encouraging comments.

Lines 47-49: See text edits.

Line 70: See text edit.

Line 262: Remove sentence “refusal of testing…”

Line 264: Study in Zambia not relevant to testing in older people – remove.

Line 269-70: Can remove sentence.

Line 248: Remove sentence beginning “Data in…”

Line 362: The statement that HIV testing uptake was low doesn’t seem justified – remove.

Line 289: Can remove sentence beginning “The low testing…”

Lines 277-284: Could be removed.

Line 252-4/Ref 27: If these data are not specific to older people, please remove.

Authors: We thank the editor for text edits. We have edited and removed the text accordingly.

Line 49: 40/567 is 7.1% - please correct. Also, how is there a confidence interval for this result.

Authors: We thank the editor for this comment. In many studies, we use a sample to estimate population parameters. Any measurement taken from the sample group therefore provides an imprecise estimate of the true population value. We provide references for the rationale for confidence interval estimation for a prevalence.

• Hazra A. Using the confidence interval confidently. J Thorac Dis. 2017;9(10):4125-4130. doi:10.21037/jtd.2017.09.14

• Department of Statisitics. Penn State Eberly College of Science. Introduction to confidence intervals [Internet]. 2020 [Accessed 30 Nov 2020]. Available from: https://online.stat.psu.edu/stat200/lesson/4/4.2

• Eayres D. Technical briefing 3 : Commonly used public health statistics and their confidence intervals [Internet]. 2008 [Accessed 30 Nov 2020]. Available from: http://www.ukiacr.org/publication/technical-briefing-3-commonly-used-public-health-statistics-and-their-confidence

Line 53: The revised sentence should probably be reworded to eg “The majority of older adults in this study accepted testing for HIV and 4.1% of those tested were newly diagnosed with HIV as a result of this test” or similar.

Authors: We thank the editor for corrections and suggestions. We have corrected the data and we have reworded the sentence in the conclusion of the abstract.

Line 72: This line is now repetitive of the sentences above. Please rephrase to state ART has contributed to the abovementioned increases in numbers of PLWH 50+.

Line 78: State the age definition of older adults in Reference 5.

Line 103: Here and elsewhere remove abbreviations where they are used only once.

Line 110-11: 20,000/120,000 does not equal 18.3%. Please check actual numbers and correct.

Line 115: States the study was carried out in Lome, but in response to reviewer 1 it is stated the study was carried out in both Lome and other health regions in Togo. Please clarify, and if only hospitals in Lome were involved amend the title as suggested by Reviewer 1.

Line 181: See comment above – it is not clear how a point prevalence has a 95% CI?

Authors: Please kindly refer to the response above. 

Line 206: p=<0.001 for both factors? Table 1 indicates other significant differences between men and women here eg monthly income, health insurance. Please mention these important differences in the text.

Line 211: Please define the Mood test in Methods.

Line 223-4: Perhaps flip this result and state that 23 (57.5%) were newly diagnosed as a result of this testing.

Table 1 and 2: Please correct age brackets to either 50-59 and ≥60 OR 50-60 and >60 (ie not 50-60 and ≥60).

Discussion:

The additional studies relevant to Africa which have been included in the revised Discussion are of interest to gain context of the data generated in the present study. However, as written this revised Discussion lacks structure and logical flow and is far too long. It would be of use to briefly mention the data pertaining to HIV prevalence/testing in older people in Africa and compare with the current findings, then discuss possible reasons and suggested interventions. Some specific suggestions include (but are not limited to):

Line 250-1: Where is the result indicating testing was not offered? State or remove sentence.

Authors: We thank the editor for this remark. This statement was based on the results presented in figure 1. We added this result in the “result section” and stated it in the “discussion section”. 

Line 274-76: Rephrase to say HIV testing campaigns targeting older people may be required.

Line 294 (and also line 337): The statement that there is limited data on HIV prevalence in older adults is inconsistent with the studies cited after, and in the Introduction, which detail a number of studies. Please rephrase. 

This section (from lines 294 to 342) needs to be significantly shortened; group similar studies from different countries together, only include data relevant to older populations, critically compare and contrast results from this study with those previously reported and avoid long lists of prevalence data with minimal interpretation.

Authors: We thank the editor for this comment. We have shortened this section of the discussion in the revised manuscript.

Lines 338 and 342 – these statements seem to contradict?

Authors: We have removed the statement in line 342.

---

## [Editor Report · Decision Letter 2]

21 Dec 2020

PONE-D-20-09735R2

HIV testing uptake and prevalence among hospitalized older adults in Togo: a cross-sectional study

PLOS ONE

Dear Dr. GBEASOR-KOMLANVI,

Many thanks for submitting your revised manuscript to PLOS ONE and for your considered attention to previous reviewer and editorial comments. We feel the manuscript is worthy of publication but still requires minor grammatical and accuracy changes to ensure it meets PLOS ONE’s publication criteria. Therefore, we invite you to submit a revised version of the manuscript that addresses the points detailed below.

We look forward to receiving your revised manuscript.

Kind regards,

Anna C Hearps

Academic Editor

PLOS ONE

Additional Editor Comments (if provided):

Keywords: “HIV uptake” should probably be “HIV testing” or “HIV testing uptake”. “Associated factors” is unlikely to be a useful search term – please remove/replace

Line 53: Remove “infected” from this sentence.

Line 109 – Should be 16.7%

Line 128 – Please provide a reference for the SHOT study if pubished.

Statistical analysis section (lines 177-): Please briefly explain in this section why Mood’s median test (which has a lower power than eg Mann-Whitney) was used. Please also explain why some categorical variables in Table 1 were assessed with Chi squared test and some with Fisher’s test. 

Line 202: Refer to Table 1 in line 202 when results first mentioned

Line 223-4: Regarding the use of the CI for the point prevalence, I thank the authors for the explanation and references provided. For the reader’s benefit please briefly explain in the text how these values were calculated and cite an appropriate reference or previous publication where you have explained this. Please also remove the CI from the abstract (line 49).

Line 229-30 – Replace male and female with males and females (plural)

Line 245-6 – Sentence is repeated in line below – please remove

Line 249 “In our study, one in four hospitalized older adults (25.7%) 250 had never been tested for HIV.” Repeats line 242-3 above. Remove 1 instance.

Line 255 Add replace “, whether” with “or whether” to make sentence grammatically correct.

Line 265: Change “health care providers feel uncomfortable…” to “health care providers **may** feel uncomfortable…”

Line 268: Add “an” before “alternative” to make sentence grammatically correct

Line 277 DHS defined in line 271 – use abbreviation

Line 298  - change to “males” (plural)

Line 330 – replace “more” with “better”
---

## [Author Response · Author response to Decision Letter 2]

5 Jan 2021

Responses to the Editor

Keywords: “HIV uptake” should probably be “HIV testing” or “HIV testing uptake”. “Associated factors” is unlikely to be a useful search term – please remove/replace

Authors: We thank the Editor for these comments. We have modified keywords in the revised manuscript.

Line 53: Remove “infected” from this sentence.

Authors: We have removed the term ‘infected’ in the revised manuscript. 

Line 109 – Should be 16.7%

Authors: We thank the Editor for this comment. We have corrected accordingly in the revised manuscript.

Line 128 – Please provide a reference for the SHOT study if published.

Authors: A more detailed description of the SHOT study has been previously published in PloS One. We provided the reference in the revised manuscript. 

25. Gbeasor-Komlanvi FA, Tchankoni MK, Bakoubayi AW, Lokossou MY, Sadio A, Zida-Compaore WIC, et al. Predictors of three-month mortality among hospitalized older adults in Togo. BMC Geriatr. 2020 Nov 26;20(1):507. doi: 10.1186/s12877-020-01907-y.

Statistical analysis section (lines 177-): Please briefly explain in this section why Mood’s median test (which has a lower power than eg Mann-Whitney) was used. Please also explain why some categorical variables in Table 1 were assessed with Chi squared test and some with Fisher’s test.

Authors: We thank the reviewer for these comments. We added an explanatory paragraph in the statistical section.

;

Line 202: Refer to Table 1 in line 202 when results first mentioned

Authors: This comment has been taken into account in the revised manuscript. 

Line 223-4: Regarding the use of the CI for the point prevalence, I thank the authors for the explanation and references provided. For the reader’s benefit please briefly explain in the text how these values were calculated and cite an appropriate reference or previous publication where you have explained this. 

Authors: This comment has been taken into account in the revised manuscript. 

Please also remove the CI from the abstract (line 49).

Authors: This comment has been taken into account in the revised manuscript. 

Line 229-30 – Replace male and female with males and females (plural)

Authors: This comment has been taken into account in the revised manuscript. 

Line 245-6 – Sentence is repeated in line below – please remove

Authors: We thank the Editor for this remark. The sentence has been removed in the revised manuscript.

Line 249 “In our study, one in four hospitalized older adults (25.7%) 250 had never been tested for HIV.” Repeats line 242-3 above. Remove 1 instance.

Authors: We thank the Editor for this remark. We removed the sentence in the revised manuscript.

Line 255 Add replace “, whether” with “or whether” to make sentence grammatically correct.

Line 265: Change “health care providers feel uncomfortable…” to “health care providers may feel uncomfortable…”

Line 268: Add “an” before “alternative” to make sentence grammatically correct.

Line 277 DHS defined in line 271 – use abbreviation

Line 298 - change to “males” (plural)

Authors: We thank the Editor for these remarks. We corrected the sentences accordingly.

Line 255 Add replace “, whether” with “or whether” to make sentence grammatically correct 

Line 265: Change “health care providers feel uncomfortable…” to “health care providers may feel uncomfortable…”

Line 268: Add “an” before “alternative” to make sentence grammatically correct.

Line 277 DHS defined in line 271 – use abbreviation

Line 298 - change to “males” (plural)

Line 298 - change to “males” (plural)

Line 330 – replace “more” with “better”

Authors: We thank the Editor for this comment. We have corrected accordingly in the revised manuscript.

---

## [Editor Report · Decision Letter 3]

15 Jan 2021

HIV testing uptake and prevalence among hospitalized older adults in Togo: a cross-sectional study

PONE-D-20-09735R3

Dear Dr. GBEASOR-KOMLANVI,

We’re pleased to inform you that your manuscript has been judged scientifically suitable for publication and will be formally accepted for publication once it meets all outstanding technical requirements.

Kind regards,

Anna C Hearps

Academic Editor

PLOS ONE
---

## [Editor Report · Acceptance letter]

21 Jan 2021

PONE-D-20-09735R3 

HIV testing uptake and prevalence among hospitalized older adults in Togo: a cross-sectional study 

Dear Dr. Gbeasor-Komlanvi:

I'm pleased to inform you that your manuscript has been deemed suitable for publication in PLOS ONE. Congratulations! Your manuscript is now with our production department. 

Kind regards, 

on behalf of

Dr. Anna C Hearps 

Academic Editor

PLOS ONE